# Geographical Patterns of Genetic Variation in Locoto Chile (*Capsicum pubescens*) in the Americas Inferred by Genome-Wide Data Analysis

**DOI:** 10.3390/plants11212911

**Published:** 2022-10-29

**Authors:** Nahuel E. Palombo, Carolina Carrizo García

**Affiliations:** 1Instituto Multidisciplinario de Biología Vegetal, Universidad Nacional de Córdoba, CONICET, Córdoba 5000, Argentina; 2Department of Botany and Biodiversity Research, University of Vienna, 1030 Vienna, Austria

**Keywords:** *Capsicum*, RAD-seq, population genomics, geographic differentiation, domestication

## Abstract

The locoto chile (*Capsicum pubescens*) is a regionally important food crop grown and marketed throughout the mid-highlands of South andCentral America, but little is known about its evolution and the diversity it harbours. An initial scan of genetic diversity and structure across its cultivation range was conducted, the first one using a genomic approach. The RAD-sequencing methodology was applied to a sampling of *C. pubescens* germplasm consisting of 67 accessions from different American countries, covering its range of distribution/cultivation on the continent. The RAD-seq SNP data obtained clustered the accessions into three major groups, with a high degree of admixture/reticulation among them. Moderate but significant differentiation and geographic structuration were found, depicting a south–north pattern in the distribution of genetic variation. The highest levels of diversity were found among central-western Bolivian individuals, while the lowest was found across Central America-Mexican germplasm. This study provides new genome-wide supported insights into the diversity and differentiation of *C. pubescens*, as well as a starting point for more efficient use of its genetic variation and germplasm conservation efforts. The findings also contribute to understanding the evolutionary history of *C. pubescens*, but further investigation is needed to disentangle its origin and diversification under domestication.

## 1. Introduction

Chile peppers (*Capsicum*, Solanaceae) are famous spices and vegetables consumed worldwide. The genus *Capsicum* comprises 43 species native to tropical and temperate regions of the Americas, with the centre of diversity in the Andes [1]. Among them, there are five domesticated species: *C. annuum*, *C. chinense*, *C. frutescens*, *C. baccatum,* and *C. pubescens*. The three economically most important species are *C. annuum*, *C. chinense,* and *C. frutescens*, which are cultivated in many countries around the globe [2,3]. The tworemaining domesticated chiles, *C. baccatum* and *C. pubescens*, are cultivated and consumed mostly in South and Central America [2,4].

The hot chile *C. pubescens* (Figure 1) is better known in Andean regions of South America as ‘locoto’ or ‘rocoto’ (from the indigenous terms *luqutu* and *rukutu*, respectively), holding a greater cultural and economic importance in the Central Andes, especially in Bolivia and Peru [1,5,6]. It is mainly grown in mid-elevation to highlands from north-western Argentina to central Mexico (Figure 1a) [1,7,8,9], where it is grown extensively in courtyards and small family plots (Figure 1b), with surplus sold in local markets. The species is clearly distinctive from the other chiles with the presence of conspicuous pubescence, large brownish-black seeds, and primarily purple flowers (Figure 1c and Appendix A) [1]. The fruits are hot fleshy berries with a broad range of variations in size, shape, and colour (Figure 1d and Appendix A) [6,7]. Although well-defined cultivars cannot be delimited, different local varieties or landraces can be informally recognised [7,10].

Ruiz and Pavon originally described *C. pubescens* in 1799 from plants cultivated in Peru [1]. Yacovleff and Herrera [11], citing evidence from the writings of colonial historians, included the species among the plants cultivated and consumed by early Peruvian peoples, from around 4,000 years before the present [12]. It has been hypothesized that its domestication took place in Bolivia and/or Peru [7,13,14,15], about 6000 years ago [10], and has been followed by human-assisted range expansion to other areas of the Americas [16,17,18]. Unlike other cultivated chiles, this species is known only as a cultigen and has not been found in the wild so far [1,6]. Fruits of smaller size occur in Bolivia, suggesting that Bolivian material is closer to the ancestral gene pool of *C. pubescens* [7]. Nonetheless, to our knowledge, no archaeological evidence is reported for Bolivia to date, so the origin and domestication history of the cultigen remains unclear. Within the genus *Capsicum*, *C. pubescens* forms a morphologically distinct group together with the wild species *C. eximium*, *C. eshbaughii* and *C. cardenasii* [7,13,17], which also occur in Bolivia and have been proposed to be its closest allies and/or putative ancestors. Recent phylogenetic studies based on genome-wide data suggest that *C. pubescens* is not an isolated lineage, but a sister species to a small clade formed by *C. eximium*, *C. eshbaughii,* and *C. cardenasii*, all four species conforming the so-called clade Pubescens [19]. Those results also revealed that none of these three wild species or any other *Capsicum* species would be the wild ancestor to *C. pubescens* [19]. Indeed, the lineage of *C. pubescens* diverged in the upper Pliocene, earlier than the clade formed by *C. eximium*, *C. eshbaughii,* and *C. cardenasii*, with these three species diversifying from the mid-Pleistocene [19].

In addition to its uncertain origin, *C. pubescens* is the less exploited among the five domesticated chiles, most likely because of its environmental requirements (i.e., a cool, freeze-free environment, long growing season, mid-high elevations) and the high fruit fleshiness, which makes them prone to rotting quickly [20]. Cultivation of *C. pubescens* outside the Americas is very limited, although it has been introduced and grown in Indonesia more than 100 years ago [21]. Nevertheless, it is becoming progressively more relevant as its market demand has increased outside the Americas due to a renewed interest in ethnic cuisines [4,22,23]. The importance of the locoto chile lies not only in the use of the fruit as a spice and vegetable but also due to its alkaloids (capsaicinoids) and carotenoids, both used in the pharmaceutical industry, medicine, agriculture, etc. [1,24].

Characterising genetic variation within crop species is key in the study of agrobiodiversity, since it may provide a useful framework for the effective use and conservation of that diversity [25,26,27]. Despite its importance as a food crop and its cultural significance, little is known about the evolution of *C. pubescens* and the levels of genetic variation and structure it harbours. Previous molecular studies based on a small number of DNA markers, such as amplified fragment length polymorphisms (AFLPs) and simple sequence repeat markers (SSRs), have provided some insight into its variability [28,29]. These studies found genetic differentiation in the species according to geographic origin/collection country but analysed a reduced sampling focused on germplasm from the Central Andes (i.e., Ecuador, Peru, and Bolivia). However, to date, there are no studies that have examined materials grown across the wide geographic range of *C. pubescens* in the Americas.

This study aimed to lay the groundwork for a genomic perspective of the variability within the locoto chile. To this end, single nucleotide polymorphism (SNPs) markers generated by Restriction-site-Associated DNA Sequencing (RAD-seq) [30] were used to explore, for the first time, the geographical distribution of genetic variation within *C. pubescens* along its range of distribution/cultivation in the Americas. As a reduced-representation sequencing technique, RAD-seq allows the identification of thousands of SNPs across the genome [30] and has been proven to be an applicable tool for investigating population structure and diversity in non-model organisms [31,32], including crop species. This approach has been applied to crops that are not widely cultivated but are important to the economies and communities of developing regions e.g., [33,34,35,36,37], as in the case of locoto chile in the Central Andes. Therefore, through the intended study, the following four questions have been addressed: (1) How much genetic diversity is present in *C. pubescens*? (2) Is the variability of the species genetically structured? (3) How is its genetic variation distributed geographically? and (4) What factors could be influencing the observed genetic variation?

## 2. Results

### 2.1. Sequencing and Assembly

A total of 112,026,584 reads were generated by Illumina sequencing of the RAD-seq libraries. After demultiplexing and filtering of low-quality reads, 103,086,817 were retained. The number of reads per individual ranged from 192,601 to 3,662,664 (average 1,393,065; standard deviation 684,492) (Appendix A). Out of the 74 genotyped individuals, seven were excluded because they represented low coverage and/or outlier samples. The final dataset holds a total of 67 individuals.

The ipyrad pipeline was run separately for datasets used in downstream analyses. For the dataset used in SplitsTree, there were 140,508 prefiltered loci and 44,507 filtered, with 183,569 SNPs and 41,636 unlinked SNPs. For the population inferences dataset, the filtered loci were 101,132 with 345,793 SNPs and 83,813 unlinked SNPs. After filtering out loci/SNPs with more than 80% missing data and a minor allele frequency of less than 0.05, the pruned dataset used consisted of 1462 unlinked biallelic SNPs.

### 2.2. Phylogenetic Network

The phylogenetic network showed a clear geographic split in *C. pubescens*, separating the southern from the northern accessions (i.e., Bolivian-Argentinian vs Mexican-Central American accessions) (Figure 2a). While there was an elevated degree of reticulation in the network, five well-defined genetic groups related to five relatively different sampling areas were distinguished, following a south–north geographic pattern. The first three groups included accessions collected from Bolivia to northern Argentina. One of them primarily contained samples from central-southern Bolivia (acquired mainly around Cochabamba and Santa Cruz de la Sierra) to northern Argentina. Another group was represented by individuals collected *in situ* in central-western Bolivia, which were located growing under natural conditions and family gardens in three localities in the surroundings of La Paz (Apa Apa, Coroico, and Huancané; ca. 100 km NW of La Paz). The third one clustered samples from Villa Serrano, a central-Bolivian small town situated in mid-elevated highlands (2000 m) about 190 km SW of Santa Cruz de la Sierra. The remaining two groups (among the five recovered) consisted of materials from Peru to Mexico. One of them encompassed samples from Peru to Ecuador, and also a single individual collected in north-western Argentina (# 270). The last group consisted of Central American and Mexican individuals, including two samples of unknown origin (# 264,267).

Ten accessions sampled from local markets in La Paz, Bolivia (# 186–189,196,197,243, 246,249,259) were located at the centre of the network together with two samples from Cusco, Peru (# 233,255) and one from Salta, Argentina (# 271) (Figure 2a). This set of 13 samples was clustered in less well-defined subgroups with relatively short splits and much reticulation corresponding to high levels of hybridization or genetic admixture, so their affinities were confusing and could not be assigned to any of the five main groups defined above.

### 2.3. Genetic Clustering

The DAPC and Admixture analyses favoured a model with *K* = 3 genetic clusters, as both the Bayesian information criterion and cross-validation test converged on this number (Figure 3a), tough additional informative structure was observed for *K* = 2, 4 and 5. The results of both clustering approaches were highly congruent (see below), mostly reflecting the genetic groups identified by the network analysis (Figure 2a). Main differences were found in the assignment of most of the individuals from La Paz markets (Bolivia), Peru, and Ecuador, which showed high levels of genetic admixture/shared ancestry (i.e., low assignment probability to a particular group). Particularly, accessions from La Paz markets were inferred to have high levels of admixture (Appendix A). In the best fit model (*K* = 3), DAPC supported three geographically structured clusters, with two discriminant functions (dimensions) describing the relationships (Figure 3b). These clusters roughly corresponded to a southern cluster composed of accessions from northern Argentina to south-central Bolivia, a (large) centre cluster comprising samples from central-western Bolivia, Peru, and Ecuador, and a northern cluster consisting of samples from Central America to Mexico, hereinafter called G1, G2 and G3, respectively. The G2 cluster also included two samples from Cusco, Peru (# 233,255) and one from Salta, Argentina (# 271) that were not assigned to any group in the network analyses (Figure 2a). The marked separation between G1 and G3 on the DAPC plots (Figure 3b) illustrates a considerable genetic difference between them.

Results of the Admixture analyses were largely correlated with the DAPC clustering but some accessions from Peru and Ecuador were assigned to G3 (Figure 3c). Samples from La Paz markets (Bolivia) showed high levels of genetic admixture from the three clusters. At *K* = 2, there was a strong differentiation between individuals from Bolivia to Argentina on the one hand, and individuals from Peru and Ecuador to Central America and Mexico, on the other (Figure 3c). A south–north pattern in the geographic distribution of genetic variability in *C. pubescens* was recognised, as in the topology of the phylogenetic network (Figure 2a). The *K* = 4, 5 sub-optimal models were also informative, showing a substructure within G2 thoroughly consistent with the groups inferred by the phylogenetic network (Figure 2a). At *K* = 4, the DAPC analyses recognized a fourth cluster composed of the set of samples from Villa Serrano (Bolivia) and, at *K* = 5, the remaining samples within G2 were split into individuals from Peru and Ecuador on the one hand and those from central-western Bolivia on the other. Differences with the Admixture results were found at *K* = 4, where individuals from central-western Bolivia and Villa Serrano were clustered together into a different group from that of the Peruvian-Ecuadorian samples (Figure 3c). Almost all of these samples showed considerable levels of genetic admixture. For *K* = 5, however, both approaches were highly congruent and it was possible to recognize the same five clusters (Figure 3c). Once again, samples from La Paz markets (Bolivia) exhibited admixture between these subclusters.

In the PCA, the first three-axis explained 20.5% of the variation in the data (Figure 2b). These results also supported a diffuse and moderate population structure, with the position of samples in the two-dimensional space being almost a function of their geographic locations (i.e., samples from nearby localities clustered together) with no well-defined breaks between clusters. Taking the *K* = 3 model described above, the PC1 differentiated G1 and G3 clusters, showing a wide overlap among G2 accessions (Figure 2b). Overall, individuals sorted in G2 were less well separated and localised at the centre of the plot (Figure 2b). Villa Serrano samples (belonging to G2) showed an isolated position in the PC1 vs PC3 plot, pointing out some degree of genetic differentiation in agreement with the clustering results at *K* = 4, 5 suboptimal models (Figure 3c).

### 2.4. Genetic Diversity and Differentiation

Following the best fit model (*K* = 3), standard measures of genetic diversity were calculated for the three main clusters of *C. pubescens* (Table 1) using the pruned dataset of unlinked biallelic SNPs. For the Admixture results, similar levels of genomic diversity were found for G1 and G2, although the latter had a higher total number of alleles, allelic and private allelic richness, and observed heterozygosity than the former. While G1 and G2 had very similar observed and expected heterozygosities, G1 showed fewer heterozygotes than expected, and this deficit was significant. A comparable pattern was obtained in the assessment of DAPC clusters (Table 1). Overall, both clusters, G1 and G2, had higher genetic diversity than G3. The calculations of genetic diversity were also performed by randomly sampling ten individuals for each cluster/group to discard the influence of sampling size (smaller in G3), with the results following the same trend (Appendix A). The same calculations were performed for the *K* = 4, 5 sub-optimal models and, again, the same pattern was recovered (Appendix A). The samples from central-western Bolivia showed the highest diversity, including allelic richness, private allelic richness, and observed heterozygosity. Overall, the genetic diversity in *C. pubescens* was higher in the Bolivian territory whereas it decreased towards the north end of its cultivation range, so the Central America-Mexico cluster (G3) was the less diverse.

Moderate genetic differentiation (F_ST_ = 0.162) and low inbreeding (F_IS_ = 0.024) across and within groups of *C. pubescens* were observed based on Admixture clustering. The DAPC results also converged into this pattern. For *K* = 3, pairs of clusters were significantly differentiated from one another by pairwise calculations of F_ST_ (Table 2), suggesting either low levels of allele sharing or differences in allele frequencies between clusters. The pairwise F_ST_ values were also consistent in showing a correlation between genetic and geographic distance. When the pairwise F_ST_ calculations were performed for the *K* = 4, 5 sub-optimal models, the results showed a similar pattern (Appendix A). To rule out the possible effect of the population size, F_ST_ comparisons among clusters of *C. pubescens* were also performed with ten individuals randomly sampled for each cluster/group and the results followed the same trend (Appendix A).

According to the Admixture clustering, the analysis of molecular variance (AMOVA) revealed a significant genetic structure between clusters/populations (*F* = 0.165; *p* = 0.001) (Table 3). The major proportion (16.5%) of the genetic variation that was not attributable to variation within individuals (which amounted to 83.9%) was partitioned among the three main clusters. Congruent results were obtained in the assessment of DAPC clusters (Table 3).

## 3. Discussion

This is the first study to evaluate genomic variation derived from NGS technologies, specifically Restriction-site-Associated DNA Sequencing (RAD-seq) methodology, for *C. pubescens*. The number of RAD-seq reads obtained and filtered, the number of loci/SNPs found and the values of the estimated parameters were consistent with previous reports for population studies in non-model plants e.g., [38,39,40] demonstrating the usefulness of the RAD-seq approach to obtain high-quality markers to perform genome-wide population inferences in *C. pubescens*. Genetically distinct clusters with different levels of genetic diversity could be defined, which correlated with the geographic origin of the accessions. The patterns of genetic diversity and structure observed within *C. pubescens* would reflect trends in human use and trading of germplasm in the Americas.

It has been proposed that *C. pubescens* holds a narrow genetic diversity as a consequence of a founder effect during its domestication [41]. The species is the lesser known and widespread among the domesticated chiles, due probably to its environmental requirements and fruit fleshiness, that make fruits difficult to store and move over long distances. Although these features might suggest that *C. pubescens* could have low genetic variability due to domestication, the species is known only as a cultigen and not from the wild, so this hypothesis has not been tested empirically. The variability within the species has been partially studied under different approaches such as morphological e.g., [7,23,42], phytochemical e.g., [24,43,44], and genetical [28,29]. These data pointed to *C. pubescens* as the domesticated chile having the narrowest genetic diversity, but there is no research undertaken on its variability across its entire cultivation range in the Americas. Even though the current work is not an exhaustive study of *C. pubescens* genetic variation, since the sampling could be more dense, it represents an initial contribution that stands out from previous work in three main points: the analyses based on genome-wide SNPs, the inclusion of a higher number of accessions to cover a broader cultivation range in America (including both latitudinal extremes), and (when possible) more detailed information of cultivation and sales locations of the studied samples. For this reason, these results consent to make some comments about what factors could be influencing the observed genetic variation in the locoto chile, as well as its origin and diversification under domestication, and the geographic distribution of diversity.

The different clustering approaches applied here resolved similar groupings, primarily identifying three geographically structured genetic groups within *C. pubescens*. Two main and clearly differentiated groups corresponded to germplasm from northern Argentina to south-central Bolivia (G1) and from Central America to Mexico (G3), respectively. A third less well-defined group was represented by accessions from central-western Bolivia, Peru to Ecuador (G2). In addition to these three clusters, the Admixture, principal components, and genetic diversity analyses showed that G2 had high levels of genetic admixture and diversity. Within this large cluster, a substructure was recognized as consistent with the phylogenetic network grouping (i.e., a total of five genetic groups), also indicating a higher admixture and genetic diversity in central-western Bolivian materials. This was in line with the elevated degree of reticulation observed in the network, pointing out considerable levels of gene flow and/or incomplete lineage sorting within G2. Conversely, the G3 showed the lowest levels of genetic diversity and high inbreeding. Taken together, these results revealed that the genetic variation within *C. pubescens* was structured although moderately differentiated, depicting a south–north pattern in the distribution of genetic diversity and lineage splitting in the species. It is worth noting that variability in the fruit features (colour, shape, fleshiness) was represented in all three groups, with the exception of individuals from La Paz surroundings (Appendix A, see below).

Patterns of genetic variation in crop species can illuminate their history of domestication and expansion [43], as has been explored in the best-known domesticated *Capsicum* e.g., [44,45]. Traditionally, crops are thought to have a centre of diversity relatively near where they were originally domesticated [46], and then experience a loss of this baseline diversity caused by bottlenecks and selection as a consequence of dispersal and introduction into new territories [47,48]. In this study, it was found a south–north geographic pattern in the distribution of genetic diversity in *C. pubescens*. Moreover, to the south, the Bolivian accessions showed greater diversity than those from other areas, while the genetic diversity decreased towards the north of the cultivation range (up to Mexico). A south–north differentiation was also reported by Ibiza et al. [28] using AFLP and SSR data, who analysed samples from Bolivia to Ecuador, being the accessions also separated according to the source country. The current data reinforce the idea of geographic diversification within the species. The presence of larger genetic diversity in Bolivia is congruent with the research of Eshbaugh [7,20], who reported greater morphological variation in this country, highlighting fruits of smaller size. Samples collected in central-western Bolivia (i.e., La Paz surroundings) showed unique genetic variation, including large private allelic richness and, more or less consistently, the highest measures of diversity. It is worth mentioning that these accessions were collected *in situ* from adult plants and their seeds were afterwards cultivated under controlled conditions, so their particular morphology was corroborated: sparse pubescence (almost absent), small flowers mostly 5-merous, and the smallest and fleshiest fruits (Appendix A). These individuals were found both under cultivation (family garden in Coroico, La Paz Department) or growing freely in disturbed habitats in the field (Apa Apa and Huancané, La Paz Department), so the latter could represent a de-domesticated or feral phenotype of *C. pubescens*. Nevertheless, these distinctive characteristics, reminiscent of a minor or incomplete domestication syndrome, and their elevated genetic diversity, suggest that this material could be closer to the ancestral gene pool of *C. pubescens*. The mid-highlands of central Bolivia have already been proposed as the hypothetical centre of origin for *C. pubescens* [7,10,13], but no wild population has been recorded to date [1]. The mentioned area is also the native habitat of two wild relatives of *C. pubescens*, namely *C. eximium* and *C. cardenasii*, which together with *C. eshbaughii* encompass the small sister clade of *C. pubescens* [19], emphasising the relevance of this region in terms of the diversity and evolution of the locoto chile and its wild allies. In fact, the Central Andes were inferred as the ancestral area of origin of the entire clade Pubescens [19]. Considering the fact that *C. pubescens* cultivation is mainly restricted to particular environmental conditions, Rick [6] suggested that perhaps the only sites in which the wild forms would grow have been occupied by humans and its cultigens, and then they might have hybridized with the improved forms, losing part of the original features and making it difficult to distinguish the wild from the cultivated locoto. The current findings strongly stress that new and extensive expeditions in central-western to north-western Bolivian highlands (specifically, the inter-Andean valleys from south-eastern La Paz to the northwest) are needed to propose better-supported hypotheses about the origin and diversification of *C. pubescens.*

The levels of genetic structure in domesticated plants are largely determined by the diversifying effect of population isolation (e.g., traditional varieties or landraces) balanced against the homogenising effect of gene flow and planting of homogeneous cultivars [48,49]. Since *C. pubescens* is cultivated mainly on a regional-local scale, the role of a large number of smallholder farmers in germplasm exchange and crossing, as well as in the local development and/or retention of traditional varieties, are factors that can explain the genetic structure and diversity patterns obtained. The high degree of admixture found in the central-western Bolivian and Peruvian samples (particularly those acquired in La Paz markets; Figure 1d) may evidence human influence in driving gene flow and thus population structure in the locoto chile. In addition, it is worth noting the differentiation of the accessions from Villa Serrano (Bolivia) as the four accessions from this locality were sorted into a distinct sub-group by the different clustering approaches employed. Villa Serrano is a small town with a long cultivation tradition of chile peppers [4,50] but it has poor connectivity with large urban centres (pers. obs.), so this finding could be understood as an event of geographic isolation and local differentiation of locoto chile genotypes.

In addition to advancing the study of the genetic variation of *C. pubescens* in its putative native range, the South American Central Andes, this work also analysed germplasm from its two latitudinal extremes of cultivation in America: northern Argentina to the south, and central Mexico to the north. The first studies concerning the variation of the species [6,7,51] did not include Argentinean materials. Nowadays, *C. pubescens* is recognised as an introduced cultivated species in Argentina [52,53], being grown in north-western areas of the country (i.e., Jujuy and Salta Provinces) [1], where marketed plants and fruits are also directly introduced from Bolivia [54] (pers. obs.). In line with these records, the results obtained here indicate that the Argentinean germplasm is mostly related to the central-south Bolivian genotypes (G1). Nevertheless, two individuals closely related to the Peruvian materials were also found, which would evidence a separate event of germplasm introduction by human trade/exchange.

Regarding the northern extreme of the *C. pubescens* cultivation range (i.e., Central America and Mexico), the accessions studied here formed a distinct genetic group (G3) with the lowest levels of genetic diversity and admixture. This would be consistent with records of the introduction of *C. pubescens* into Central American and Mexican highlands occurring in the 20th century [13,55,56] rather than the development of historical cultivars. In fact, no indigenous names are reported for the species in this region, while it is popularly known by common names in Spanish, such as ‘chile manzano’, ‘perón’, ‘canario’ or ‘caballo’, that refer to fruit shapes and colours, to the sensation caused by capsaicinoids in humans or to a particular use in the cuisine [1]. Based on the samples analysed, it can be suggested that the introduction of *C. pubescens* to Central America would have originated from Peruvian-Ecuadorian sources due to the observed larger affinity. Silvar and García-González [29] also found two Central American samples clustered within Peruvian germplasm using SSR data. The measures of genetic diversity showed that G3 was the least variable group, which may indicate that a bottleneck or founder effect may have occurred during the introduction of the locoto to the north of the continent, concurring with the suggestion of diversity loss due to selection and introduction of a crop into new regions [46].

Understanding how evolutionary processes (e.g., gene flow, selection, diversification, adaptation) impact crop genetic diversity and characterising the standing genetic variation is key to germplasm conservation and crop improvement efforts. The three (to five) genetic groups of *C. pubescens* inferred here following a genome-wide approach represent a useful starting point for understanding the extent and the partitioning of genetic variation across its cultivation range in the Americas. Trends in human use and trading would be key factors that have shaped the patterns observed in this work. The finding of greater genetic diversity in central-western Bolivia and a diversity loss associated with its dispersion towards the north of the continent provides some clues for new studies that seek to unravel the domestication history of the locoto. Of particular value might be an enlargement of the Bolivia to Peru sampling (i.e., La Paz surroundings to the northwest) to provide better coverage of its putative native range. In this context, to better understand the origin and domestication of *C. pubescens*, its relationship with its wild sister species should be analysed in-depth.

## 4. Materials and Methods

### 4.1. Sampling

In order to perform a first scan of the geographic distribution of genetic diversity in *C. pubescens* along its cultivation range, a total of 74 accessions from several South–Central American countries were included in the analyses (Figure 1a): Argentina (10), Bolivia (39), Peru (10), Ecuador (3), Costa Rica (2), Guatemala (2), Mexico (5), unknown (3) (Appendix A). All materials were acquired in local markets or germplasm banks, exceptionally from individuals collected *in situ* under natural conditions and in family gardens in Bolivia (Appendix A). It was attempted to sample the broad morphological and geographical variation of locoto fruits (colour, shape, size), emphasizing Bolivian sources (Appendix A). Adult plants were grown from seeds of the sampled fruits at the Instituto Multidisciplinario de Biología Vegetal (IMBIV, Cordoba, Argentina) or in the Botanical Garden of the University of Vienna (HBV, Vienna, Austria).

### 4.2. DNA Isolation and RAD-Seq Library Preparation

Genomic DNA was isolated using either the DNeasy Plant Mini^®^ kit (Qiagen, Germantown, MD, USA) or the Invisorb^®^ Spin Plant Mini Kit (Invitek Molecular GmbH, Berlin, Germany) from leaves dried in silica gel following the manufacturers’ instructions. DNA extracts were purified using the NucleoSpin^®^ gDNA Clean-up kit (Macherey-Nagel GmbH & Co., Düren, Germany). DNA was checked for quality by agarose gel electrophoresis and quantified using a Qubit^®^ three Fluorometer (Invitrogen, Waltham, MA, USA).

RAD-seq libraries preparation followed the protocol of Paun et al. [57] with modified settings for DNA fragmentation (i.e., six cycles 90” off, 60” on) and using 150 ng for each sample. A two-index combinatorial approach was followed using standard Illumina indexes and inline barcodes. Multiplexed libraries were sequenced (single-end) using an Illumina HiSeq 100 bp System at the Vienna Biocenter Core Facilities [58]. Raw data quality was assessed with FastQC v.0.11.9 [59]. Raw reads were demultiplexed using illumina2bam [60] and process_radtags in Stacks v.2.41 [61] with simultaneous sequence quality filtering (minimum *Phred* scores set to 20 and allowing a single mismatch in the barcodes). After demultiplexing reads within each sublibrary based on the individual barcode, six samples were excluded based on low sequencing coverage; additionally, one sample was removed from the dataset after preliminary analysis showed it to be an outlier.

### 4.3. Loci Assembly and SNP Calling

Variable SNP loci from all samples were filtered and clustered de novo in ipyrad v0.9.82 [62] using default options for diploids except for the parameter *clust_threshold* (i.e., the level of sequence similarity at which two sequences are identified as being homologous, and thus cluster together). Clustering threshold (ct) selection approaches aim at determining appropriate ct values to establish homology while avoiding clustering of paralogous RAD-seq loci [63]. Application of such a strategy is highly popular to reduce the risk of introducing assembly error to the dataset e.g., [63,64,65,66,67], ensuring the assembly of homologous loci and maximizing sequence variation. To do so, a ct range of 0.85–0.99 (in 0.01 increments) was tested and assembly results were plotted (Appendix A). Based on this result, the consensus ct was set to 0.925.

To produce a dataset for phylogenetic analysis, which can tolerate relatively large amounts of missing data, filtering was conducted using the parameter *min_sample_locus* = 34 (i.e., minimum number of samples per locus), that is 50–51% of samples. Because population genetic analyses are less tolerant of missing data and rare alleles than phylogenetic analysis, additional filtering was performed to obtain the appropriate datasets for the execution of the structure and diversity analyses. First, demultiplexed reads were filtered in ipyrad using the parameter *min_sample_locus* = 4. The *.vcf file obtained was then processed in VCFtools v.0.1.15 [68]. Only biallelic SNPs with a minor allele frequency above 0.05 and genotyped in at least 80% of the individuals were kept. In the final step, the SNPs dataset was pruned to one SNP per locus to generate a dataset of unlinked markers using vcf_parser.py script [69].

### 4.4. Phylogenetic Analysis

To explore the phylogenetic relationships and distances between *C. pubescens* accessions, an unrooted phylogenetic network was constructed using SplitsTree v.4.17 [70]. The program PGDSpider v.2.1.1.5 [71] was used to convert the complete *usnps.phy file from ipyrad (41,636 unlinked SNPs) to nexus format, which was used as input. The split network was inferred with the uncorrectedP method, which ignores ambiguous sites, and the Neighbor-Net algorithm. This method uses aspects of Neighbor-Joining [72] and SplitsTree to create a network that allows visualising multiple hypotheses simultaneously.

### 4.5. Detection of SNPs Putatively under Selection

The occurrence of outlier/non-neutral loci, interpreted as candidates for loci under selection, was assessed for the pruned dataset (1462 unlinked biallelic SNPs) using Bayescan v.2.1 [73] and a Principal Component Analysis (PCA) approach implemented via the PCAdapt package v.4.3.3 [74] in R [75]. Only sites identified by both methods were considered true outliers. BayeScan (*K* = 2–5, 20 pilot runs and 5000 generations, with 5000 initial generations discarded as burn-in) identified zero outliers compared to 74 outliers detected by PCAdapt (*K* = 1–20, minimum allele frequency= 0.05, five PCs retained). As no outliers were common across both approaches, those identified by PCAdapt were considered false positives, therefore the analyses described below were conducted using all loci.

### 4.6. Population Structure Analysis

First, a discriminant analysis of principal components (DAPC) [76] was implemented to infer genetic clusters using the pruned dataset (1462 unlinked biallelic SNPs) and the function dapc of the adegenet package v.2.1.3 [77] in R [75]. The DAPC was executed using *K*-means clustering to identify the optimal number of clusters from *K*= 1–10, and the appropriate number of clusters was then inferred using Bayesian information criterion (BIC). The cross-validation function Xval.dapc was used to determine the optimal number of PCs to be retained. Second, the genetic structure was also inferred using Admixture v.1.3 [78], as an alternative clustering approach to get a better picture of genetic admixture or shared ancestry within the accessions. The *.vcf file was converted to ped and map files using VCFtools v.0.1.15 [67], then further converted to bed, bim, and bam files using PLINK v1.90b6.24 [79]. The analyses were run for *K* = 1–10 with 100 replicate runs per K. The most informative value of *K* was explored using the cross-validation (CV) method and results were plotted using R. Barplots indicating the genetic group assignments were drawn in R. Third, a principal component analysis (PCA) was also conducted in the R package adegenet to visualize samples in two-dimensional genetic space.

### 4.7. Genetic Diversity and Differentiation

Once genetic structure was inferred, summary population genetic statistics were calculated separately in two subsets based on each clustering approach: (a) for clusters inferred using DAPC, and (b) Admixture clusters with mixed ancestry individuals removed (<70% membership probability) to avoid effects of potential genetic admixture or gene flow effects. Common measures of genetic diversity were calculated for the inferred clusters of *C. pubescens* accessions using the pruned dataset of unlinked SNPs. Total number of alleles (A), observed heterozygosity (H_O_), and gene diversity (H_E_, the expected heterozygosity within subpopulations assuming Hardy–Weinberg Equilibrium) were calculated with the adegenet package v.2.1.3 [77] in R [75]. Allelic richness (A_R_; rarefied to account for population size) and private allelic richness (A_P_) were estimated in ADZE v.1.0 [80]. The inbreeding coefficient (F_IS_) was calculated in hierfstat v.0.5–7 [81] in R.

To test for significant genetic differentiation between the inferred *C. pubescens* clusters, pairwise values of population differentiation (F_ST_ of Weir & Cockerham) [82] were calculated via the wc function of the hierfstat package in R. The confidence intervals were generated using 1000 bootstrap replicates. Finally, testing how genetic diversity was structured within the species was conducted using analysis of molecular variance (AMOVA) [83] with the poppr.amova function of the poppr v.2.8.6 package [84] in R, with 10,000 permutations to test for significant differences. AMOVA was performed for the following levels: among genetic clusters, among individuals (within each genetic cluster), and within individuals.

## Figures and Tables

**Figure 1 plants-11-02911-f001:**
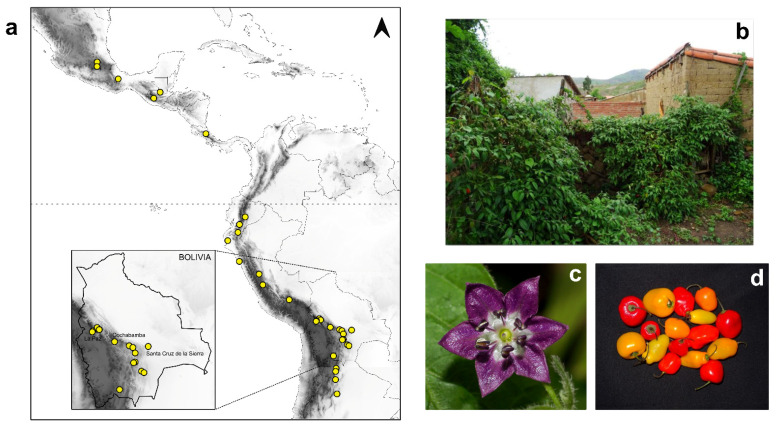
Geographic distribution and featuring of *C. pubescens*. (**a**) Area of *C. pubescens* cultivation in South and Central America. Points on the map indicate the location of the studied samples. A detail of the sampling in Bolivia is shown (lower left corner) and major cities are labelled. (**b**) Large shrub growing in family garden in Moro Moro, Bolivia. (**c**) Flower and (**d**) Fruits acquired in street markets in La Paz, Bolivia.

**Figure 2 plants-11-02911-f002:**
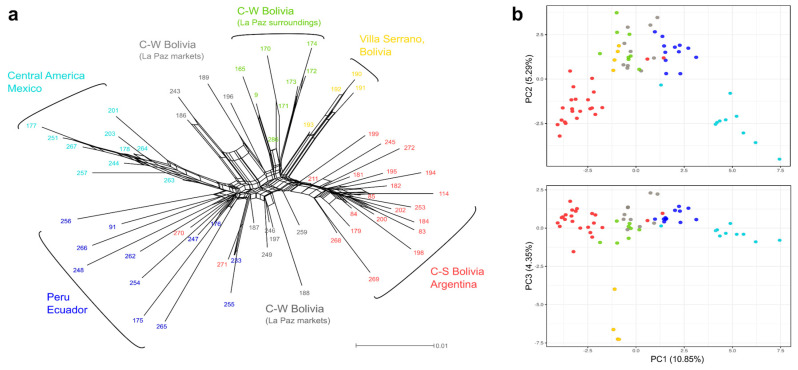
Phylogenetic network and Principal component analyses of 67 *C. pubescens* samples using 41,636 unlinked SNPs markers. (**a**) Neighbor-Net constructed by SplitsTree. The five genetic groups identified are shown. Samples are labelled with different colours according to their origin/country. (**b**) Principal component analysis (PCA). Axes are labelled with the percentage of variation explained by the two PCs (PC 1 vs. 2 and PC 1 vs. 3). Samples/dots are coloured according to phylogenetic network grouping.

**Figure 3 plants-11-02911-f003:**
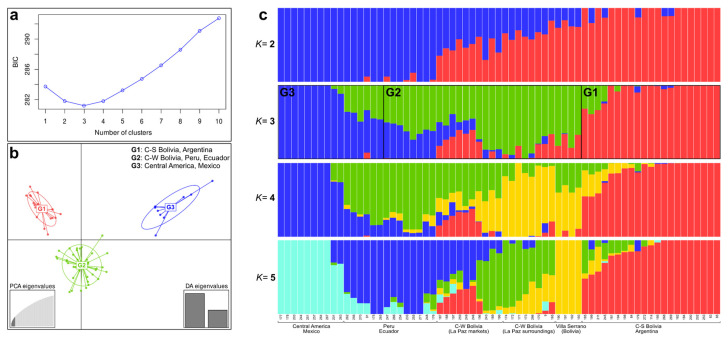
Genetic clustering and differentiation of 67 *C. pubescens* samples using 1462 unlinked biallelic SNPs. (**a**) Bayesian information criterion (BIC) values for different number of clusters (*K*). (**b**) Discriminant analysis of principal components (DAPC) at *K* = 3. Six PCs and two discriminant eigenvalues were retained during analyses to describe the relationships among the clusters. Each circle represents a genetic cluster, and each dot represents a sample. The different colours represent the three clusters. (**c**) Cluster assignments for *K* = 2–5 estimated in Admixture. Each bar represents a sample and the colours represent partitioning of the sample genotype in each group. The accessions are sorted according to phylogenetic network grouping (Figure 2a). G, genetic cluster/group.

**Table 1 plants-11-02911-t001:** Common measures of genetic diversity for 67 *C. pubescens* samples from the three inferred clusters (*K* = 3) using 1462 unlinked biallelic SNPs.

Cluster	N	%P	A	A_R_	A_P_	H_O_	H_E_	F_IS_	F_IS_ (95% CI)
**Admixture**									
G1	18	90.17	2624	1.394	0.159	0.201	0.190	−0.018	−0.045, −0.010
G2	14	94.40	2747	1.492	0.213	0.245	0.232	−0.007	−0.024, 0.012
G3	10	70.24	2044	1.228	0.075	0.102	0.108	0.101	0.080, 0.170
**DAPC**									
G1	21	93.43	2732	1.413	0.158	0.212	0.199	−0.015	−0.051, −0.020
G2	37	98.8	2889	1.444	0.176	0.195	0.211	0.082	0.059, 0.150
G3	9	69.46	2031	1.267	0.0801	0.101	0.104	0.084	0.079, 0.111

N = number of individuals; %P = percentage of polymorphic SNPs; A = total number of alleles; A_R_ = allelic richness, A_P_ = private allelic richness; H_O_ = observed heterozygosity; H_E_ = expected heterozygosity; F_IS_ = inbreeding coefficient; F_IS_ (95% CI) = lower and upper 95% confidence intervals of inbreeding coefficients.

**Table 2 plants-11-02911-t002:** Pairwise genetic differentiation (F_ST_) among the three inferred genetic clusters (*K* = 3) of *C. pubescens* using 1462 unlinked biallelic SNPs. F_ST_ values are given below the diagonal. Lower and upper limits of 95% confidence intervals are given above the diagonal. G, genetic cluster/group.

**Admixture**	G1	G2	G3
G1	-	0.086–0.100	0.233–0.267
G2	0.093	-	0.158–0.186
G3	0.250	0.172	-

**DAPC**	G1	G2	G3
G1	-	0.068–0.082	0.223–0.257
G2	0.076	-	0.117–0.143
G3	0.240	0.130	-

**Table 3 plants-11-02911-t003:** Analysis of molecular variance (AMOVA) for 67 *C. pubescens* samples from the three inferred clusters (*K* = 3) using 1462 unlinked biallelic SNPs.

Level	% Variation	*F*-Value	*p*-Value
**Admixture**			
Among clusters	16.529	0.165	0.001 *
Among individuals within clusters	−0.518	−0.006	0.586
Within samples	83.989	0.160	0.001 *
Total	100		
**DAPC**			
Among clusters	12.171	0.122	0.001 *
Among individuals within clusters	3.979	0.045	0.028
Within individuals	83.850	0.162	0.001 *
Total	100		

* Significant at *p* ≤ 0.05.

## Data Availability

The datasets generated for this study are available on request to the corresponding authors.

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
