# Peer review of "Geographical Patterns of Genetic Variation in Locoto Chile (Capsicum pubescens) in the Americas Inferred by Genome-Wide Data Analysis"

_plants, 2022, doi:10.3390/plants11212911_

Round 1
Reviewer 1 Report
The work by Palombo and Garcia describes the evaluation results of genomic variations obtained with NGS technologies, in particular RAD-seq, for Capsicum pubescens.
In my opinion, the work is well written, the results of the analyzes are sufficiently documented.
The results presented in the work provide a lot of information, from the level of genomic analysis, regarding the diversity and differentiation of C. pubescens.
The authors of the paper ask themselves 4 main questions about genetic diversity in C. pubescens, the genetic structure of variation and its geographical distribution, and finally the question about the factors that may influence the observed genetic variability in C. pubescens.
In the work, I found answers to the first three questions, but I did not find an answer to the last question.
The most important advantages of the work include: demonstration of the usefulness of the RAD-seq technique for obtaining high quality markers, identification of the 3 main clusters within C. pubescens and consideration of the original place of domestication of C. pubescens.
I believe that the work is interesting and important in terms of the subject matter.
Author Response
We are thankful for the comments and suggestions received about our work.
In relation to the four questions that were addressed in our work, the following comment was made: "In the work, I found answers to the first three questions, but I did not find an answer to the last question". The topic referred to in the fourth question was addressed in several passages of the discussion but perhaps in a somewhat vague way. Therefore, we have emphasized the ideas on the matter, as can be seen in lines 292-295 and 405-406.
Please see the new version manuscript attached.

Reviewer 2 Report
The manuscript by Palombo and García entitled “Geographical Patterns of Genetic Variation in Locoto Chile (Capsicum pubescens) in the Americas Inferred by Genome wide Data Analysis” addressed four important questions based on the sequence analysis of 67 accessions from different American countries. The authors claim that ‘provides new genome-wide supported insights into the diversity and differentiation of C. pubescens, as well as a starting point for more efficient use of its genetic variation and germplasm conservation efforts’. The work is technically sound piece of research, and within the scope of Plants, it may need a minor revision before its acceptance for publication.
1. Provide more references in the Introduction, for example, ‘Genetic diversity, population structure, and relationships in a collection of pepper (Capsicum spp.) landraces from the Spanish centre of diversity revealed by genotyping-by-sequencing (GBS)’
Author Response
We are thankful for the comments and suggestions received about our work.
We are aware that there are numerous interesting works referring to the best-known Capsicum species, but we did not seek to go further in that sense. Our work focuses on a poorly known domesticated species of the genus, which in turn belongs to a phylogenetic lineage different from those of the better-studied species. Nevertheless, we have added a new citation of the work mentioned by the reviewer, which would fit in the Discussion section. This can be seen in lines 314-316. The reference has been added in the References section (number 44).
Please see the new version manuscript attached.
